# I'm InterPlanetary, Get Me Out of Here!
# Accessing IPFS From Restrictive Environments

Leonhard Balduf*
Technical University of Darmstadt
Darmstadt, Germany
leonhard.balduf@tu-darmstadt.de

Sebastian Rust*
Technical University of Darmstadt
Darmstadt, Germany
sebastian.rust@tu-darmstadt.de

Björn Scheuermann
Technical University of Darmstadt
Darmstadt, Germany
scheuermann@kom.tu-darmstadt.de

## ABSTRACT

We investigate how well IPFS functions in real-world restrictive network environments. In a series of experiments, we utilize four vantage points, one of which lies behind the Great Firewall of China (GFW), to ascertain how various parts of the IPFS ecosystem perform in these settings. We test HTTP gateways and find that, although they are not systematically blocked, only about a third function in China, in comparison to Germany. Evaluating P2P functionality, we run experiments on data exchange between the four nodes. We find that the GFW has little measurable impact on these functionalities. The main inhibiting factor for P2P functionality remains network address translation (NAT). Lastly, to help NATed nodes spread their content, we propose and evaluate using public gateways as temporary replicators, but find only marginal gains.

## CCS CONCEPTS

• **Networks** → **Network measurement**; **Peer-to-peer networks**; *Firewalls*; *Network reliability*.

## KEYWORDS

peer to peer, network address translation, firewall, networks, china

**ACM Reference Format:**
Leonhard Balduf, Sebastian Rust, and Björn Scheuermann. 2023. I'm Inter-Planetary, Get Me Out of Here! Accessing IPFS From Restrictive Environments. In *4th International Workshop on Distributed Infrastructure for the Common Good (DICG '23), December 11–15, 2023, Bologna, Italy*. ACM, New York, NY, USA, 6 pages. https://doi.org/10.1145/3631310.3633487

## 1 INTRODUCTION

The Interplanetary Filesystem (IPFS) [6] is one of the largest peer-to-peer (P2P) filesystems currently in operation. One of the main properties of IPFS is that content is stored decentralized, which can make it more resistant to censorship than centralized counterparts. Content is addressed by its cryptographic hash, which makes it immutable and verifiable. These properties are used by

*Both authors contributed equally to this work.

various decentralized applications [14], which enable free and uncensored exchange of information such as social networking and discussion [13], but also data storage [15], and messaging [21], Resistance to censorship is particularly important in countries with restrictive internet policies and is a prerequisite for the free flow of information and participation.

The IPFS network currently contains a steady number of $\approx 30,000$ [23] online nodes, spread across 2,700 autonomous systems and 152 countries, according to a recent study [22]. Support for accessing IPFS has further been integrated into Hypertext Transfer Protocol (HTTP) gateways (*e.g.*, Cloudflare) and mainstream browsers such as Opera and Brave, allowing easy uptake.

*Contributions.* We investigate functionality of IPFS from different real-world network scenarios, and their impact on performance. We present a comprehensive measurement study, covering the entire lifecycle of a typical IPFS operation, including an inquiry into HTTP gateways, acquisition of the IPFS client software, and finally content exchange via the IPFS P2P network. To help users in NATed or restricted environments in making their content available in the network, we propose to use community-provided HTTP gateways as data replicators, and evaluate the benefits. In designing and executing our methodology, we put special emphasis on reproducibility. We publish the entire, documented set of tools to run and evaluate our experiments[1], making it easy to replicate our work.

*Findings.* Overall, our main findings include: (1) Public gateways are not systematically blocked in China. Still, only a handful of them work. (2) It is possible, albeit not trivial, to acquire the IPFS software in this environment. (3) The Great Firewall of China (GFW) does not seem to have measurable impact on the ability of IPFS nodes to exchange data.

## 2 BACKGROUND

*IPFS.* IPFS is a content-centric network where nodes are identified via their *peer ID*, which is derived from the public key of a unique key pair. Each node advertises a set of network endpoints describing their IP address, transport protocol and port number. Content is identified by a content identifier (CID), which is a hash of the content itself, so $CID(d) = h(d)$ for a cryptographic hash function $h$ and some content $d$.[2] CIDs are location-agnostic, immutable, and not human-readable, facilitating content deduplication, data retrieval from nearby sources, and data integrity maintenance. However, before downloading content, a CID must be resolved to its providers using a Kademlia [17] distributed hash table (DHT) and

---

[1] See https://github.com/mrd0ll4r/dicg_2023_ipfs_paper_code.
[2] In practice, the CIDs include some metadata and are encoded using a self-describing format. We refer the readers to other studies [22] for a more detailed description.

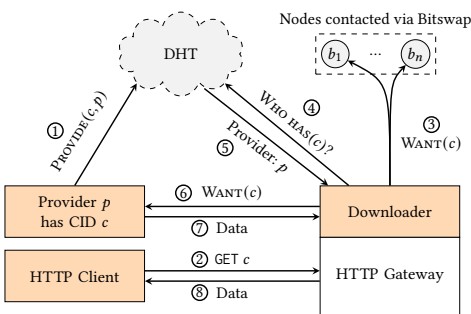

**Figure 1: Illustration of Typical Interactions in IPFS.**

the Bitswap protocol. Figure 1 depicts these network interactions, which we will now elaborate on.

*DHT.* IPFS uses a Kademlia [17] DHT implementing a key-value store. A new participant node joins the IPFS network by contacting one of the configurable bootstrap nodes. This bootstrap node provides the new node with some initial peers allowing it to fill its routing table. Recent versions of the software distinguish between DHT *servers* and DHT *clients*. Clients solely use the DHT for content resolution and routing, services provided by DHT servers. For a node to qualify as a DHT server, the software checks if it's directly accessible from the internet and not, *e.g.*, behind NAT.

*Content Advertisement.* When a user wants to add content to the network, it calculates the corresponding CID and announces itself as a provider of that CID to the DHT (cf. Figure 1 ①).[3] A provider without a public IP address (*e.g.*, a NATed node) cannot directly receive download requests for the content it provides, unless it is already connected to the downloader via Bitswap. Generally, NATed nodes first establish a connection to a random DHT server supporting the relay protocol that will act as a reverse proxy and NAT-hole-punching introducer [19]. The provider includes the IP address of the relay in the provider records it generates.

*Content Retrieval.* Downloading a data item $d$ with CID $c$ is a two-step process: (1) Providers for $c$ are found (cf. Figure 1 ③, ④, ⑤), (2) Connections to the providers are established and $d$ is downloaded from them directly via Bitswap (⑥, ⑦). The search for providers begins with a local, 1-hop broadcast via Bitswap (③) to all connected neighbours looking for the target CID. Searching via Bitswap is fast, but does not provide reliable content resolution. If unsuccessful, the downloader looks for providers of $c$ in the DHT, then simultaneously opens Bitswap connections to these providers to obtain the content.

*HTTP Gateways.* The user interacts with the network either directly via a locally running daemon, or via an HTTP–IPFS *gateway*, which translates HTTP operations to their IPFS counterparts and executes them on the daemon. *Public* gateways enable IPFS-agnostic users to access the content (cf. Figure 1 ②, ⑧) [2–4].

When a gateway receives a request for a CID, it (1) checks its local cache (2) finds and downloads the content using IPFS, and (3) returns the content to the client using HTTP. Protocol Labs maintains a list of public gateways [12].

---

[3]We refer to [3, 4, 22] for a more detailed description of the processes involved.

*Great Firewall of China.* The GFW is a multi-layered censorship and surveillance system operated by the Chinese government [24]. It is designed to block access to undesired websites and services, and to monitor and control the flow of information within the country. Prominent examples of blocked services include Google, Facebook, Twitter, and Wikipedia, as well as many foreign news outlets and human rights organizations [9]. The GFW is implemented as a combination of technologies, including IP blocking, DNS poisoning, and Deep Packet Inspection (DPI), as well as active probing and filtering of network traffic [24].

## 3 RELATED WORK

*IPFS.* We add to a growing body of research on IPFS [6]. Daniel and Tschorsch [7] provide a comprehensive overview of the IPFS ecosystem and its components, focusing on network participants and their churn. In contrast, we focus on the availability of the network itself. Trautwein et al. [22] give an overview of the functionality of IPFS and measure its client population over an extended period of time. With respect to geolocation, they find that around 24% of all nodes are located in China, making it the second most popular country for IPFS nodes. However, they also find that China has the highest share of nodes that are not reachable. In our own works, we study IPFS using DHT crawling [10], passive monitoring [3], as well as centralization in multiple parts of the ecosystem [4]. Censorship and availability of data in IPFS has been studied in the context of eclipse attacks [20].

*Internet Censorship.* Internet censorship is extensively researched and continuously monitored. Non-profit projects like OONI [18] provide tools and data to continuously monitor the scope of censorship measures. However, they do not provide any data on IPFS, but focus on the availability of centralized websites and services. FreedomHouse [1] and the OpenNet Initiative [11] provide reports on the state of internet freedom. They observe that internet freedom has been declining for the last couple of years, making censorship a pressing issue. Ensafi et al. [8] and Wu et al. [24] examine how the GFW actively probes for and blocks protocols used to circumvent censorship. These studies show that the GFW is capable of blocking undesired protocols, despite their use of encryption and obfuscation. While most of these studies focus on the availability of centralized services, we focus on the availability of decentralized services, namely IPFS.

## 4 METHODOLOGY

Our overall goal is to ascertain how easy it is to access IPFS in the context of different network environments. Among these, we evaluate (non-)NATed environments, as well as a client behind the GFW, as an example of an especially restrictive environment. To this end, we set up four vantage points: (1) A non-NATed machine in Germany, the *DE Server*, (2) A non-NATed machine in the US, the *US Server*, (3) A NATed machine in the US, the *US Client*, and (4) A NATed machine in mainland China, the *CN Client*.

The two non-NATed servers act as controls for how well nodes should be able to provide and retrieve content. The NATed machines are exemplary of home users. The US client acts as a control to the NATed machine in China: Using it, we can try to measure the additional impact of the GFW on the Chinese client in comparison

to just NAT on the American machine. Our node in China is situated in Qingdao. We confirm that blocked websites such as `google.com` are inaccessible, *i.e.*, internet access is censored.

## 4.1 Accessing Content via Public Gateways

As detailed in Sec. 2, IPFS can be accessed in various ways, with public gateways among the most common [2, 4, 22]. We evaluate the accessibility of these gateways from all vantage points by using each public gateway to fetching a widely replicated text file not included with IPFS distributions. This approach determines if gateways retrieve content from the network, use a whitelist, etc. Content validity is confirmed using SHA256 hashes, with results discussed in Sec. 5.1.

## 4.2 Content Distribution via the P2P Network

Public gateways are centralized infrastructure, the reliance upon which makes blocks easier to implement. As such, we examine content distribution via the P2P network next. We want to evaluate how well the vantage points can share data between each other directly. For this, we generate data items on the machines, add them to local IPFS nodes, and download them at 5m intervals from the other vantage points. To this end, we firstly install the IPFS client software on the four vantage points, reporting on ease of installation.

We then generate a number of data items on each vantage point. For this, and because IPFS advertises itself as a solution for the distributed web [16], we generate file sizes following a distribution of files in the web as laid out in [5]. We assume an upload bandwidth of 1 Mbps to be available on each node, and limit the file size such that every download can finish in the allotted 5m time slot. On each node, we create files with random binary content, which ensures that each data item is *unique* in the network, *i.e.*, our node is its only provider. We then add the files to the IPFS client running on the respective node. Notably, this does *not* cause the data items to be sent to the network yet, but merely generates CIDs and advertises the storing node as a provider for them (cf. Sec. 2).

Next, we generate assignments between the vantage points to download data items off each other in rounds. We achieve this by creating a random order of the four vantage points and having each node download from the next one in this order. This ensures that each is downloading from a different node Additionally, we enforce that no two nodes are simultaneously downloading from each other, which would require only one connection attempt to succeed. We randomize the order to reduce effects of a node downloading off of the same remote multiple times in sequence, which could cause connections to be reused, which in turn could affect the outcome of the experiment. We control for this in a second experiment. An example schedule is shown in Table 1.

Finally, we initiate a download according to aforementioned assignments at every time slot, for a total of seven days, which generates $\approx$ 2000 data points per vantage point. For this, the machines' clocks are synchronized using NTP, and a script which initiates the downloads according to the schedule is executed regularly via cron. We use the IPFS CLI to impose a timeout of five minutes on the operation, and record the actual time taken. We compare the SHA256 hash of the downloaded item to the expected value derived

from the original data item, and consider a download successful if the hashes match. We present the results of this experiment in Sec. 5.2.

**Table 1: Experiment Setup: Time Slots and Download Assignments. A data item $d_{n,k}$ is the $k$-th unique data item provided by node $n$.**

| Time Slot | Download Assigned to Node | | | |
|---:|:---:|:---:|:---:|---:|
| | $n_1$ | $n_2$ | $n_3$ | $n_4$ |
| 1 | $d_{n_3,1}$ | $d_{n_1,1}$ | $d_{n_4,1}$ | $d_{n_2,1}$ |
| 2 | $d_{n_4,2}$ | $d_{n_3,2}$ | $d_{n_2,2}$ | $d_{n_1,2}$ |
| | | ... | | |
| $t$ | $d_{n_4,t}$ | $d_{n_3,t}$ | $d_{n_1,t}$ | $d_{n_2,t}$ |

## 4.3 Improving Replication via Gateway Pseudo-Pinning

Content is available in the network only while at least one provider is *reachable* (cf. Sec. 2). Thus, one way for NATed or otherwise restricted nodes to increase availability of their content is through increasing its replication. To this end, some entities provide paid *pinning* services: They guarantee that content uploaded to them stays available in the network. As a cheaper alternative, albeit without any guarantees, we attempt to use public gateways as pseudo-pinning services. As laid out in Sec. 2, these gateways, like any IPFS node, cache downloaded content locally and advertise as a content provider. We thus posit that gateways can act as temporary enhancers for content availability. However, the ethical implications of using public gateways this way are discussed in appendix A.

We repeat the P2P content retrieval experiment from before, extending it in the following way: Having identified a set of globally functional public gateways via Sec. 4.1, We schedule a regular task on each vantage point to request content from the local node via two gateways every 12 hours.

Additionally, we gather peer lists from each vantage point before initiating a download. This helps us to determine if the observed results are mainly due to pseudo-pinning or node connectivity. We present the results of this experiment in Sec. 5.3.

## 5 RESULTS

## 5.1 Gateways

We evaluated public gateway availability and functionality from our four vantage points on 2023-09-18. We tested all 81 endpoints on the public gateway list[4] [12]. The results of this experiment are presented in Table 2a. Overall, 14 were functioning correctly for the well-connected systems in Germany and the US. One of these gateways was unreachable from the client in the US, which we attribute to flakiness. On the other hand, just 5 gateways were functioning correctly for the Chinese client. The gateway list is community-maintained and may have outdated entries, which contributes to the low success rate overall. However, there is a notable difference between the Chinese vantage point and the others. It is important to note that the number of functional gateways does not

---

[4]Which is published on GitHub, the availability of which we discuss in Sec. 5.2.

affect the health of IPFS overall, as they primarily facilitate external access.

We checked whether the gateways are located in China by resolving their IP addresses from within China. However, all the gateways had IP addresses outside of China, confirming that their operators do not host endpoints within China. This means connections to these gateways have to traverse the GFW. While some gateways are inaccessible in China, there does not seem to be systematic blocking of all public gateways, which would be easy to achieve using the public list.

Overall, we identify 5 gateways, listed in Table 2b, that were reachable and functional from all of our vantage points. Notably, even though one of the gateways operated by Protocol Labs (`dweb.link`) was reachable, the other one (`ipfs.io`) was not. Also, all gateways provided by Cloudflare were unreachable for the Chinese client. This hints at collateral damage, rather than targeted blocking of IPFS-related resources.

### Table 2: Gateway Connectivity Results

**(a) Functionality by Vantage Point.**

| Machine | Tested | Working |
| --- | --- | --- |
| DE Server | 81 | 14 |
| CN Client | 81 | 5 |
| US Client | 81 | 14 |
| US Server | 81 | 13 |

**(b) Functional Gateways.**

| Gateway TLD |
| --- |
| dweb.link |
| gateway.pinata.cloud |
| hardbin.com |
| nftstorage.link |
| w3s.link |

## 5.2 P2P Content Distribution

*IPFS Client.* The IPFS client software is distributed via four primary means: (1) via `dist.ipfs.tech`, (2) via `github.com`, (3) as a docker container via, *e.g.*, `docker.io`, and (4) via IPFS itself, via the InterPlanetary Name System (IPNS) name `dist.ipfs.tech`. These options all work well in Germany and the US. GitHub has a complicated history in China, and has been blocked in the past. The Chinese vantage point unsurprisingly encountered some troubles downloading off GitHub, however, at the time of writing, was able to complete the download. Interestingly, `ipfs.tech` is blocked, but `dist.ipfs.tech` is not, which could make it difficult to discover the latter. Overall, the GFW does not impose insurmountable restrictions on obtaining the IPFS client software. Even though we chose a centralized instance as a source, it is commendable that there is a decentralized offering in the distributions provided via IPFS itself. After initialization, all nodes were able to join the network via bootstrap nodes as usual.

*Content Distribution.* We performed the first content distribution experiment from 2023-09-09 to 2023-09-16 for a total of 7 days. During this, each node attempted to download 2016 data items off the other nodes, for a total of 8064 attempted downloads. Of those, 5712 ($\approx$ 71%) were successful. The results of this for each downloader/provider pair are shown in Fig. 2, left facet.

Per downloading machine, the downloads were successful in $\approx$ 58% of the cases for the DE server, $\approx$ 66% for the US server, $\approx$ 80% for the US client, and $\approx$ 80% for the CN client. The lower success rate for the two servers can be explained through the experimental setup: Each server, at each time slot, downloads off of one out of

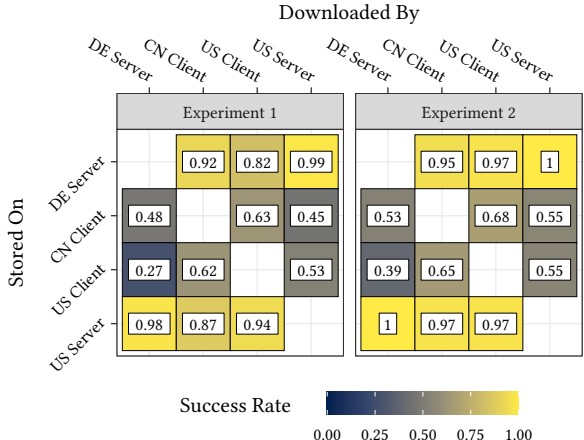

Figure 2: Download Success Rate by Source and Target, Comparison Between Experiments.

three possible remotes, two of which are NATed clients. Conversely, clients more frequently download from non-NATed servers. Notably, success rates for downloading content are similar for NATed nodes in the US and China. This leads us to believe that the GFW has no measurable impact on *outgoing* connections in IPFS, at least in this limited setting of just four vantage points. It is important to note that we just investigate a scenario where a node downloads from just one other node. In a real-world scenario, a node would likely download from multiple nodes, increasing the chances of success.

Examining the same data through the lens of the *storing* machines, we see that $\approx$ 91% of the data stored on the DE server was downloaded successfully, $\approx$ 93% for the US server, $\approx$ 47% for the US client, and $\approx$ 52% for the CN client. This is in line with the previous observations: A node is much more likely to *provide* content, if it is *not* NATed. Just like before, we can conclude that the GFW does not appear to affect incoming connections in IPFS during this experiment.

The impact of NAT on P2P systems has been noted numerous times. IPFS itself even ships with a NAT hole-punching mechanism [19], which is enabled by default. We are positively surprised that even NATed nodes are able to provide their content in $\approx$ 50% of the cases. Examining at the success rates per downloader/provider pair as shown in Fig. 2, however, we can see that downloads *between* the NATed machines are far more likely to succeed than from one of the non-NATed machines. We suspect this is because the NATed machines have longer-lived connections in general, and between each other in particular. We posit this is because they do not function as DHT servers (cf. Sec. 2), and thus see less churn. We test this hypothesis in the next experiment.

## 5.3 Gateway Pseudo-Pinning

For the second experiment, we reset the nodes, generate 2000 fresh data items, and set up the gateway cache refresh scripts as laid out in Sec. 4.3. We utilize the 5 gateways that were reachable from all vantage points, as determined in Sec. 5.1.

We performed the second experiment from 2023-09-18 to 2023-09-25 for a total duration of 7 days. During this, each node attempted

to download 2000 data items off the other nodes, for a total of 8000 attempted downloads. Of those, 6148 ($\approx$ 77%) were successful.

Per downloading machine, the downloads were successful in $\approx$ 64% of the cases for the DE server, $\approx$ 70% for the US server, $\approx$ 87% for the US client, and $\approx$ 86% for the CN client. Examining the same data through the lens of the *storing* machines, we see that $\approx$ 97% of the data stored on the DE server was downloaded successfully, $\approx$ 98% for the US server, $\approx$ 53% for the US client, and $\approx$ 59% for the CN client.

Overall, we conclude that success rates in the second experiment are slightly higher than the earlier experiment. Next, we will investigate whether the effects of this can be attributed to gateway pseudo-pinning, or connectivity between the nodes.

**Table 3: Download Success Rate by Downloading Machine.**

| | Experiment 1 | | | Experiment 2 | | |
|---|---|---|---|---|---|---|
| Downloaded By | $n$ | Succ. | Rate | $n$ | Succ. | Rate |
| DE Server | 2016 | 1160 | 0.58 | 2000 | 1283 | 0.64 |
| CN Client | 2016 | 1621 | 0.80 | 2000 | 1716 | 0.86 |
| US Client | 2016 | 1608 | 0.80 | 2000 | 1748 | 0.87 |
| US Server | 2016 | 1323 | 0.66 | 2000 | 1401 | 0.70 |

**Table 4: Download Success Rate by Storing Machine.**

| | Experiment 1 | | | Experiment 2 | | |
|---|---|---|---|---|---|---|
| Stored On | $n$ | Succ. | Rate | $n$ | Succ. | Rate |
| DE Server | 2016 | 1834 | 0.91 | 2000 | 1948 | 0.97 |
| CN Client | 2016 | 1049 | 0.52 | 2000 | 1174 | 0.59 |
| US Client | 2016 | 956 | 0.47 | 2000 | 1064 | 0.53 |
| US Server | 2016 | 1873 | 0.93 | 2000 | 1962 | 0.98 |

*Effects of Node Interconnectivity.* As outlined in Sec. 2, the IPFS software attempts to hold a number of connections at any point in time. For the version v0.22.0 we utilized, this is a value $32 < x < 96$. In our second experiment, the clients in the US and China averaged 50 to 65 connections, respectively, placing them well within the target. The servers in Germany and the US, however, held an average of 230 and 168 connections, significantly exceeding the *HighWater* mark. This eventually leads to the connection manager closing connections. We suspect this higher number of open connections is due to their roles as DHT servers, leading other nodes to contact them frequently.

We check whether connections to our other vantage points are present in a node's peer list before attempting a download. The results of this are shown in Fig. 3. Both servers and clients frequently maintain connections within each group, which would allow them to discover and download off of each other immediately. This is not the case between servers and clients. Comparing this to the download success rates presented in Fig. 2, we can see a relation: The two clients can effectively exchange data, but success rates drop noticeably when servers download from clients. We suspect the same situation presented itself in the first experiment.

In general, we thus conclude that the results obtained in Sec. 5.3 are dominated by node interconnectivity and ease (or difficulty) of establishing connections, especially to NATed nodes. Gateway

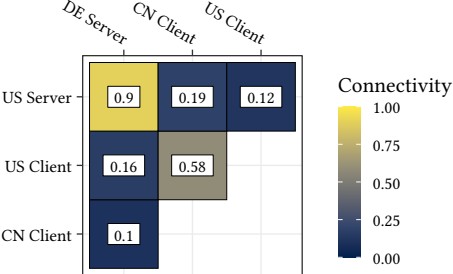

**Figure 3: Connectivity Between Nodes, Measured as Fraction of Presence in Peer Lists.**

pseudo-pinning potentially had an impact, however, within the single-digit percentages. For future work, one might try to refresh gateway caches more often, or utilize more than two gateways per item.

## 6 CONCLUSION AND DISCUSSION

In this work we examined accessibility of IPFS, a censorship-resistant hypermedia protocol, from within restrictive network environments. We perform tests on four vantage points in the US, Germany, and China. For that, we first test the availability of public HTTP gateways and find that only 5 are accessible from within China. Even though the majority of gateways are unavailable, it seems unlikely that IPFS gateways are systematically blocked by the GFW, since this would be trivial to implement via the public gateway list. Next, we experimentally test whether IPFS can traverse the GFW, both for outgoing and incoming connections. We find no measurable impact of the GFW on the functionality of IPFS as a P2P system. Rather, we find that a deciding factor of how well a node can provide content is in whether it is NATed, even though IPFS uses a NAT-traversal mechanism [19]. Finally, we attempt to improve content availability by using public gateways as pseudo-pinning services, with only moderate success.

It seems that, at this moment, IPFS is functional even in restrictive environments, such as China. This benefits the decentralized, censorship-resistant ecosystem. However, it remains uncertain if IPFS can evade the authorities' attention indefinitely. Central points of failure, such as public gateways, software distribution mechanisms, or bootstrap nodes, are attractive targets to cripple functionality of the otherwise decentralized IPFS.

For future work, we would like to add a non-NATed vantage point in mainland China. This would allow us to have a control to the non-NATed systems outside the GFW. Additionally, vantage points in other networks with tight authoritative control, such as Iran or Pakistan, would be helpful for a comprehensive analysis. In this context, an investigation into the *content* hosted on IPFS, in particular w.r.t. material censored in any of these jurisdictions, would be interesting.

## ACKNOWLEDGMENTS

The authors would like to thank the anonymous referees for their valuable comments and helpful suggestions. This work was supported by the German Research Foundation (DFG) within the Collaborative Research Center (CRC) SFB 1053: MAKI (https://gepris.dfg.de/gepris/projekt/210487104).

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

# A  ETHICS

In the pseudo-pinning experiment, we use 2000 data items per vantage point, making 4000 requests per vantage point, split across 5 functional gateways. Each gateway receives an average of 800 requests twice a day from four vantage points, totaling 6400 requests per day per gateway. In [22], the authors present a dataset of one instance of a public gateway, which received 7.1M requests per day. As such, the impact of our experiments is limited. Considering the limited benefit observed in our experiments, however, we cannot generally recommend using gateways as pinning services.

The client in China was rented by the authors from a commercial provider without involvement of a third party.