# OpenReview forum: "I'm InterPlanetary, Get Me Out of Here! Accessing IPFS From Restrictive Environments"
_ACM.org/Middleware/Workshop/DICG — DICG 2023_

### Official Review · Reviewer_CeRv · 2023-10-27
**An interesting experimental paper, with some early results on the impact of the GFW (Great Firewall of China) on IPFS**

**Rating:** 8
**Confidence:** 4

**Review:**

The paper presents an experimental study of the impact of the GFW  (Great Firewall of China) on IPFS. The study is preliminary and only involves four nodes: 2 server nodes with front-end addresses (in Germany and the US), and two NATed clients (in the US and China). The results show that the GFW does impede gateway availability in IPFS, but probably because of collateral damages rather than because of a targeted decision to hamper IPFS. Connectivity experiments show that being located behind a NAT (whether in the US or China) has a strong impact on the ability of nodes to provide content (as expected), but that the situation of the NATed machine in China is not that different from that of the NATed machine in the US. Overall, I found the paper pleasant to read, informative, and likely to lead to interesting discussions at the workshop, hence my score.

### Some questions/remarks:
- The authors have used a machine in China to assess the behavior of a governmental censorship infrastructure (GFW). From an ethical viewpoint, could their research lead to reprisals against any Chinese colleague (if any) who helped set up this research? If yes, was this cleared through some appropriate ethical board? It might be good to comment on these issues in the final version of the paper. (As reprisals have occurred in the past in similar situations.)
- You could possibly replace yellow with green in Figures 2 and 3 and also represent values by filling each square more or less, while keeping the color (which would make it easier to compare actual values, as color shades can be hard to gauge).

---

### Official Review · Reviewer_SkQF · 2023-10-30
**Good and interesting paper with detailled experiments**

**Rating:** 7
**Confidence:** 4

**Review:**

This paper studies the connectivity of IPFS, e.g., the ability to exchange data, in restricted environments such as China. The authors setup a series of experiments, involving four vantage points spread across the world, with and without NAT restrictions. One of the main conclusions is that China's firewall has negligible impact on the main functioning of IPFS.

Overall, I enjoyed reading the paper. The authors present various experiments and thought carefully about their setup. I think the topic of this paper (measurements in deployed P2P networks) very much fits the scope of the workshop as well.

A suggestion for improvement would be to better discuss the implications of particular findings. Even though I'm aware of the page limit, the lack of a more thorough discussion on particular findings or statements made it difficult to have a more extensive picture of the work. For example, according to Table 1, only 14 out of 81 gateways are working from the US and Germany. This is a pretty low fraction and I would be interested in knowing more about the implications of this. The same applies for the connectivity of gateways from China - are 5 gateways enough for to let the P2P network run in a sustainable manner? What is the minimum number gateways required for decent performance and reliability? Giving a sound answer to these questions probably requires additional experiments but I would appreciate some thoughts on these manners.

The authors mention other related work on IPFS as well as censorship but do not provide a description on how this work differs from state-of-the-art research in censorship of P2P networks.

Given the above points, I recommend to accept the paper.

Other comments:
- Please convert Figures 2 and 3 to vector-based images so their quality does not degrade when scaling/print out the paper.
- The authors mention that the decentralized nature of IPFS makes it resistant to censorship. Please note that decentralization does not directly result in censorship resistance but instead makes it more difficult for attackers to mount privacy attacks.
- For reproducibility, the authors promise to open-source the tools and scripts but didn't provide an URL. Please make sure to make these artefacts public and findable.
- Near the end of Section 5.3, the authors note that the two servers cannot download off each other relatively well but according to Figure 2, this does not seem the case; the download success rate between the servers seems to be pretty high and according to Figure 2, the connectivity between servers is high as well.

---

### Official Review · Reviewer_1tkK · 2023-10-30
**The exchange success rate among four asymmetric IPFS peers is measured with two experiments running a week each and also covering the case of a IPFS client having to traverse the Chinese firewall. Success can be as low as 27% and public HTTP gateways do not help much in availability.**

**Rating:** 6
**Confidence:** 3

**Review:**

The paper lays out a measurement campaign regarding the ease of content exchange on top of IPFS. Two servers and two clients, all strategically positioned, are "cross-connected" and tested against each other.

The experiment setup is convincing and well described, I liked the combinatorial approach of mixing all roles (producer/consumer, client/server). The promise of "covering the entire lifecycle of a typical IPFS operation" including SW acquisition is somehow held, but little is said about the end systems (OS, performance class, storage capacity, observed disk and network load except the # of connections). The list of references contains many very fresh publications. The main insight is captured in the two sentences in the middle of the paper: "we thus conclude that the results obtained in Sec. 5.3 are dominated by node interconnectivity and ease (or difficulty) of establishing connections, especially to NATed nodes. Gateway pseudo-pinning potentially had an impact, however, within the single-digit percentages."

On a critical node: The insight remains thin because the outcome is not discussed much e.g., is a 62% download success rate good, in some metrics? Relatedly, in light of a decentralized mindset, could local/regional content accessibility weight more than the implicit "global 100% success rate" target (it would be good to make this explicit)? For example, IPFS islands could serve a community well, even if restricted to inside Iran. The brittleness of bootstrap information is mentioned but the text fatalistically stops there without further discussion or hints to mitigation possibilities.

On the technical level I could not expand how you compute and impose the content generation and consumption schedule, from the sentence "This is done by constructing a graph of the four vantage points and permuting possible cycles in it.[..] according to aforementioned assignments..." The "graph" is just a fully meshed network hence you can discard this fact, while you are interested in all non-intersecting paths of length 3, arranged in time, for which it suffices to generate the 4! permutations of the node vector, right? How do you then orchestrate this i.e., some content flows only via the nodes 4->2->1->3: if some content generation happend at time t, is node 2 waiting until t+d, node 1 waits until t+2d etc? Do all nodes have week-long respective scripts?

At the editing level I did not get the title's "get me out of here" as the work focuses on "getting in" (availability). There are several self-citation entries  (2,3,4,8,11) in the related work section where it's not clear in which way they relate to this paper, except the self-citation - please spell out exactly which result you rely on, or contrast with, or remove them.

Nits:
- check the sentence with "while at least one provider and reachable" ('is' instead of 'and'?)